# Neuroimmune Crossroads: The Interplay of the Enteric Nervous System and Intestinal Macrophages in Gut Homeostasis and Disease

**DOI:** 10.3390/biom14091103

**Published:** 2024-09-02

**Authors:** Meng Lou, Robert O. Heuckeroth, Naomi E. Butler Tjaden

**Affiliations:** 1Department of Pediatrics, The Children’s Hospital of Philadelphia Research Institute, Abramson Research Center and Department of Pediatrics, Pearlman School of Medicine at the University of Pennsylvania, 3615 Civic Center Blvd, Philadelphia, PA 19004, USA; heuckerothr@chop.edu (R.O.H.); tjadenn@chop.edu (N.E.B.T.); 2Division of Gastroenterology, Nutrition and Hepatology, The Children’s Hospital of Philadelphia, 3401 Civic Center Blvd, Philadelphia, PA 19004, USA

**Keywords:** neuroimmune, macrophage, enteric nervous system, gut inflammation

## Abstract

A defining unique characteristic of the gut immune system is its ability to respond effectively to foreign pathogens while mitigating unnecessary inflammation. Intestinal macrophages serve as the cornerstone of this balancing act, acting uniquely as both the sword and shield in the gut microenvironment. The GI tract is densely innervated by the enteric nervous system (ENS), the intrinsic nervous system of the gut. Recent advances in sequencing technology have increasingly suggested neuroimmune crosstalk as a critical component for homeostasis both within the gut and in other tissues. Here, we systematically review the ENS–macrophage axis. We focus on the pertinent molecules produced by the ENS, spotlight the mechanistic contributions of intestinal macrophages to gut homeostasis and inflammation, and discuss both existing and potential strategies that intestinal macrophages use to integrate signals from the ENS. This review aims to elucidate the complex molecular basis governing ENS–macrophage signaling, highlighting their cooperative roles in sustaining intestinal health and immune equilibrium.

## 1. Introduction

A hallmark of gut mucosal immunity is the delicate balance between defense against pathogenic microbes and immunological tolerance of both the commensal microbiome and benign food byproducts. Constitutively exposed to the “external environment”, immune cells of the gut face the unique challenge of distinguishing friend from foe in the context of a non-sterile environment. How does the gut immune system manage to consistently protect the host while minimizing collateral damage to the epithelial monolayer? The solution involves a dynamic system integrating facets of the immune system, the enteric nervous system, the epithelium, and microbes.

The gastrointestinal tract is the largest immunological organ of the body, harboring up to 70% of lymphocytes alongside an expansive network of macrophages in both the lamina propria (LP) and organized mucosal-associated lymphoid tissue (MALT) [1,2]. From RORγt^+^ regulatory T-cells (Treg), important for commensal tolerance, to intraepithelial lymphocytes, crucial for epithelial integrity, to CD103^+^ dendritic cells (DC), essential in promoting Treg differentiation, gut immune cells are endowed with tissue-unique characteristics that protect against excessive inflammation. These unique characteristics differentiate these gut-resident cells from immune cells in other tissues de-spite their shared hematopoietic ontogeny. This niche-dependent functional pleomorphism is demonstrated beautifully by a recent single-cell RNA sequencing study, which reveals striking heterogeneity of immune cells across 20 diverse human tissues [3]. How gut-specific characteristics of immunity are primed to behave so heterogeneously remains an active area of study.

Gut immunity is modulated by a constellation of both intrinsic and extrinsic factors. The gut microbiome is instrumental in shaping gut immunity, particularly in the colon where more than 10^14^ bacteria reside [4]. Gnotobiotic and germ-free studies alike demonstrate the role of the microbiome in longitudinal shaping of immune responses [5]. This host–microbe interaction is further complicated by the direct and indirect effects of dietary metabolites. We recommend elegant reviews by Wiertsema et al. [6], Zheng et al. [7], and Alexander and Turnbaugh [8] on this subject, which explore the intricate intersection between immunity, microbiome, and diet in the context of human disease. Despite these contributions, crosstalk between local neurons and immune cells in the gut has yet to be fully explored.

The nervous system and immune system exhibit striking parallels: both are designed to monitor and respond to stimuli, albeit on different time scales. Whereas neurons respond on the order of milliseconds, immune circuits are primed on the order of minutes to hours and days, yet both are capable of generating long-lasting memory. Recent evidence suggests immunological memory is tightly intertwined with neuronal memory since an alleviation of inflammation is observed upon the inhibition of neurons in the insular cortex that were activated by prior infection [9]. Similarly, studies involving acute and chronic stress in the context of inflammation further underscore the importance of neuroimmune communication in the gut [10,11]. Advances in sequencing technology over the last decade suggest that the gut nervous system has a bidirectional role in gut immunity, particularly the enteric nervous system (ENS) [12,13]. The sequencing of the human ENS at the single-cell level revealed a compelling potential for communication with immune cells through released factors as well as the expression of receptors for soluble mediators of immunity [14].

Macrophages (MΦs) are highly abundant mononuclear phagocytes critical to bowel motility, gut fluid secretion, and effective host defense. These intestinal macrophages are remarkably heterogeneous in both spatial localization and function. Strategically positioned at the interface between the gastrointestinal tract, luminal bacteria, and food metabolites, lamina propria macrophages (LpMΦs) serve critical roles as first-line defenders and can activate adaptive immunity [15,16]. Aside from sampling luminal antigens from pathogenic and commensal microbes, intestinal macrophages differ from other organ macrophages because they typically reduce inflammation and promote gut homeostasis [17]. Studies in humans have shown that despite their potent bactericidal nature, intestinal macrophages are strikingly immunologically quiescent compared to the macrophages of other tissues [18]. Chronic inflammatory diseases of the gut often display features of macrophage dysfunction, suggesting their role in both intestinal repair and damage in the context of disease [19,20]. Therefore, understanding which components of the gut microenvironment bestow anergy to resident macrophages and how this immunological quiescence is perturbed in disease remains a pivotal and unresolved question.

In this review, we explore gut neuroimmune interactions in the context of homeostasis (i.e., health), inflammation, and disease. While extrinsic neurons and other immune cells play a significant role in the gut neuroimmune crosstalk (recently reviewed in Wallrapp and Chiu [21]), we spotlight the intrinsic ENS and resident gut macrophages. We provide an overview of the ENS, distilling its complexity into key points relevant to neuroimmune interactions, followed by a dive into macrophages and known crosstalk with neurons and neuronal factors, before concluding with a discussion of the implications of the ENS-MΦ axis on both developmental biology and human disease.

## 2. The Enteric Nervous System (ENS): The Second Brain

The ENS contains over 500 million neurons and glial cells in the human gut, subdivided into two plexuses: the myenteric plexus and the submucosal plexus [22,23,24]. Situated between the longitudinal and circular smooth muscle, a combination of excitatory and inhibitory neurons of the myenteric plexus coordinate and generate complex motility patterns, while submucosal plexus fibers dive towards the lumen, placing them in proximity to both epithelium and the lamina propria [23] (Figure 1). Circuitry varies drastically throughout the rostral–caudal axis, and even within the colon itself [25]. The anatomical architecture of the ENS is beautifully detailed in reviews by Newgreen and Young, Hao and Young, and Furness et al. [24,26,27]. The molecular mechanisms of development of the ENS are further illustrated in Lake and Heuckeroth [28].

Enteric neurons release a diverse collection of neurotransmitters and neuropeptides capable of shaping immunity. Recent RNA profiling studies of the ENS defined over 20 transcriptionally distinct neuron classes [29,30,31]. Classically, major neurotransmitters of the ENS include excitatory molecules such as adenosine triphosphate (ATP), serotonin (5-HT), substance P, glutamate, and acetylcholine, and inhibitory molecules such as vasoactive intestinal peptides (VIPs), somatostatin, dynorphin, and enkephalin, as well as nitric oxide (NO) and calcitonin gene-related peptides (CGRPs) [32]. Importantly, neurons of the ENS often produce more than one neurotransmitter. These molecules in the gut microenvironment can have diverse effects based on both the target cells and their source of synthesis. For example, VIP acts as excitatory neurotransmitters on neurons but consistently serves as an inhibitory mediator of circular smooth muscle contraction [33]. Likewise, 5-HT from enteric neurons is protective and promotes neurogenesis in the setting of inflammation, while mucosal 5-HT derived from enterochromaffin cells (ECs) is pro-inflammatory. This is evidenced by the amelioration of DSS colitis severity upon the knockout of TPH1 (tryptophan hydroxylase specific to ECs) which is required for 5-HT synthesis in epithelial ECs [34,35,36]. Studies over the last decade have highlighted remarkable interactions between these neurotransmitters and components of mucosal immune defense. For example, neuromedin U (NMU) signaling promotes the migration of type 2 lymphocytes in vitro, bolsters their IL-5 and IL-13 release, and acts as a potent activation signal for eosinophil degranulation [37,38]. Talbot et al. elegantly demonstrated that VIP-ergic neurons, upon host food consumption, stimulate innate lymphoid cells (ILC3) via VIPR2 to release IL-22 critical for shaping the microbiome and fat absorption [39].

In addition to classical neurotransmitters and neuropeptides, the ENS also synthesizes cytokines. For example, the ENS secretes IL-6, a molecule crucial for inhibiting the differentiation of Rorγt^+^ Tregs critical for microbiome regulation in the gut [40,41]. The neuron-specific ablation of IL-6 has been demonstrated to increase Rorγt^+^ Tregs in vivo [42]. These studies may provide insight into how the ENS can transcriptionally respond to microbes to regulate Treg populations which in turn may be able to reciprocally tune surrounding microbes [43]. Whether constituents of the ENS can directly influence the microbiome is still under investigation. Whole-mount confocal microscopy and FISH studies also reveal the ENS as a source of IL-18, which promotes goblet cell production of antimicrobial peptides critical for host defense [44]. Selective ablation of IL-18 from the ENS alone (but not immune or epithelial cells) confers increased susceptibility to subsequent infection by *Salmonella typhi* [44]. This finding implicates the ENS as a direct contributor to gut mucosal defense against infections. Finally, neurons of the ENS express receptors for both cytokines TNFα as well as TGFβ, with the latter also acting as a ligand produced by the ENS itself, opening more possibilities for neuro-immune interactions [14,45].

An improvement in sequencing technologies has allowed us to better understand the constellation of synthesized molecules and complementary receptors derived from the ENS. Even so, how these receptors and molecules derived from unique ENS neuronal subtypes work in concert to modulate the local immune system, particularly intestinal macrophages, remains largely underexplored.

## 3. Intestinal Macrophages: Sentinels of the Gut

MΦs are highly heterogeneous phagocytes capable of adapting to local environments of tissue-specific niches. MΦs are organized in a dense network in all layers of the intestine, each endowed with a specific function depending on their anatomical position (Figure 1). For example, subepithelial macrophages, classically defined as Cx3cr1^hi^MHCII^hi^F4/80^+^ in mice and CD14^+^CD11b^+^Cd11c^-^ in humans, specialize in phagocytosis and their role in promoting oral tolerance by presenting antigens to CD103^+^ dendritic cells (DCs), which then migrate to the mesenteric lymph nodes (mLNs) to induce the differentiation of tolerogenic regulatory T-cells. In contrast, neuron-associated macrophages that reside near the submucosal and myenteric plexus of the ENS have been implicated in the promotion of neuronal health in both a developmental and infectious context [46,47]. The CD169^+^ MΦs surrounding the intestinal vasculature are critical for blood vessel integrity and preventing luminal microbes from infiltrating the bloodstream [48]. Recent transcriptional analyses of gut MΦs at a single-cell level allude to the diversity and complexity of MΦ subpopulations in the intestinal niche [48,49].

The specific framework of the intestinal environment that instills MΦ phenotypes remains an active field of study. Understanding the ontogeny of intestinal MΦs in parallel to the developing ENS provides insights into their functional specialization. MΦs in the gut are broadly classified into bone-marrow-derived or embryonic self-renewing types. Self-renewing MΦs are embryonically derived from the yolk sac and the fetal liver. Unlike resident macrophages of most tissues, intestinal MΦs exist as an ontological exception; the majority of intestinal MΦs are constitutively replaced by circulating monocytes [50,51]. Fate-mapping studies in mice have shown that while fetal-liver-derived monocytes initially seed the intestines from embryonic day 8.5 (E8.5) onwards, and most tissue-resident MΦs do not persist into adulthood [52]. Indeed, these cells are replaced by hematopoietic-derived counterparts, a phenomenon dependent on the commensal microbiome, as turnover coincides with the weaning period and is greatly diminished in germ-free mice [52]. Conversely, turnover is speculated to increase in the context of inflammation, a concept supported by the perseverance of embryonically derived microglia in the brain (an immunologically quiescent tissue) compared to the gut (under chronic physiological “inflammation”) [53]. A small subset of intestinal MΦs, particularly neuron- and vessel-associated MΦs further away from the epithelial lining, are self-replicating and essential for supporting the ENS and vasculature, as a loss of these MΦs induces dysmotility and vascular leakage, respectively [48]. The nuances of the diverse subsets of intestinal MΦs are comprehensively detailed in Hegarty et al. [20] and Viola and Boeckxstaens [54].

Functionally, intestinal MΦs are central figures in epithelial integrity. MΦ-derived IL-6 regulates claudin-2 expression in gut epithelial cells—this tuning of tight junction permeability implicates intestinal MΦs in ion transport, fluid secretion, and pathogen defense [55]. Intestinal MΦs further contribute to the epithelial barrier by producing prostaglandin E2 (PGE2), a molecule important for the maintenance of the epithelial stem cell niche [56]. Depletion of intestinal MΦs unsurprisingly decreases leucine-rich repeat-containing receptor 5 (Lgr5^+^) intestinal stem cells [57]. Finally, IL-10 released by MΦs in the gut mediates epithelial repair via the secretion of the pro-repair molecule WNT1-inducible signaling protein 1 (WISP1) [58].

The MΦ differentiation paradigm is often simplified into M1 versus M2 populations. M1-activated MΦs produce pro-inflammatory cytokines and are associated with inflammation (TNFα, IL-1β, IL-6, and nitric oxide), whereas anti-inflammatory M2 MΦs produce factors aligned with tissue remodeling and wound healing (IL-10, Arg1, and YM1) [59,60,61]. Despite this simplified binary framework, it is important to acknowledge that macrophages exist in a dynamic, multi-dimensional spectrum and are prone to phenotypic switching [62,63,64]. Inflamed intestines are generally characterized by predominant M1 polarization [65], and inflammatory bowel disease (IBD) models display a combinatorial increase in pro-inflammatory M1 MΦs with a decrease in anti-inflammatory M2 MΦs [65,66]. In addition to increased M1 MΦs, these MΦs in IBD patients acquire a more aggressive phenotype, releasing more TNFα [67]. A similar pattern is observed in necrotizing enterocolitis (NEC), a life-threatening intestinal inflammation affecting premature infants. Inhibiting M1 and promoting M2 MΦ polarization decreases the incidence of NEC [68,69]. The depletion of intestinal MΦs also confers impaired vascular development in infants, a known risk factor for the development of NEC [70]. These studies demonstrate that intestinal MΦs play a decisive role in bowel inflammation and highlight the importance of investigating these cells in the context of human intestinal diseases.


*Neuron–Macrophage Interactions*


The development of the intestinal macrophage network occurs symbiotically alongside the ENS (Figure 2). Neuron-associated macrophages play a critical role in the pruning of the ENS. Ex vivo live imaging shows dynamic interactions between ENS neurons and muscularis neuron-associated macrophages, which exhibit engulfed neuronal debris in phagosomes [46]. Ablation of these macrophages during early development increases ENS synaptic density and subsequently delays bowel transit time, implying that MΦs play roles in ENS maturation [46]. Further evidence of macrophages contributing to bowel motility comes from studies on gastroparesis [71,72]. Diabetic mice lacking macrophages are protected against the development of delayed gastric emptying [73]. Additionally, bone morphogenetic protein 2 (BMP2) secreted by neuron-associated macrophages is sensed by the ENS, which then changes coordinated smooth muscle contractions, thereby altering bowel motility [74]. In return, the adult myenteric plexus serves as the primary source of colony stimulating factor 1 (CSF1), a crucial growth factor for MΦ differentiation and maintenance [74]. This reciprocal ENS–macrophage communication illustrates the codependency of these two systems during homeostasis: muscularis MΦs prune the ENS, while the ENS nurtures the muscularis MΦs. Whether hematopoietically derived lamina propria MΦs are governed by this bidirectional relationship remains to be elucidated. Additionally, prenatal *Ret^−/−^* mice, whose small intestine and colon are devoid of an ENS (i.e., total intestinal aganglionosis), surprisingly exhibit normal muscularis MΦ patterning and function, suggesting a temporal (i.e., prenatal) exception to this crosstalk [75]. This phenomenon is explained by alternative sources of CSF1, such as endothelial cells and the interstitial cells of Cajal [75]. The implications of this bidirectional crosstalk in other diseases are complexities still under investigation.

In the context of enteric infections, muscularis macrophages in the colon release protective polyamines to ameliorate infection-induced ENS loss. Polyamines (such as spermine) inhibit inflammasome activation and promote neuron regeneration [76,77]. This mechanism to preserve neurons of the ENS is driven by adrenergic receptor β2-AR signaling [78]. Anti-CSFR1 antibody-mediated depletion of MΦs ablates this protection, resulting in exacerbated neuronal loss [78]. Glia-macrophage interactions are also relevant in enteric inflammation. Damage-activated enteric glial cells release CSF1 and CCR2 to recruit anti-inflammatory MΦs to promote tissue healing and resolve inflammation [79]. This is further supported by the observation that conditioned media of glial cell cultures upregulate classic anti-inflammatory genes (*Arg1*, *Il10*, *Mrc1*) in bone-marrow-derived macrophages (BMDM), predominantly because of glia-derived CSF1 [79]. On the other hand, during chronic psychological stress, elevated levels of circulating glucocorticoids activate a subset of ENS glial cells to release CSF1 to mediate TNFα-driven enteric inflammation [10]. How these opposing CSF1 effects are integrated into intestinal MΦ signaling remains to be answered.

Investigations into the effect of common neurotransmitters released by the ENS (see above) on gut macrophages shed light on the intriguing potential of ENS-MΦ crosstalk (Figure 2). Serotonin (5-HT) is sensed by lamina propria macrophages, which express complementary receptors HTR2A and HTR3A [56]. The binding of 5-HT triggers Wnt/β-catenin signaling, stimulating prostaglandin PGE2 release by intestinal macrophages, a factor instrumental to the renewal of Lgr5^+^ intestinal stem cells [56]. Similarly, preoperative treatment with prucalopride (a 5-HT4 receptor agonist), in addition to promoting regeneration of colonic ENS neurons, also decreased MΦ-driven postoperative ileus by downregulating MΦ *Il6* and *Il8* expression [80,81]. Blocking CGRP signaling delays recovery of DSS-induced colitis in murine models, while increased CGRP levels are correlated with increased anti-inflammatory CD4^+^Tim4^+^ MΦs [82]. Daily intraperitoneal injection of CGRP subsequently increased TGFβ production by murine intestinal MΦs in vivo [82]. Finally, VIP has been demonstrated to polarize MΦs to an anti-tumor phenotype in a colon cancer model [83]. Utilizing an antagonist of VIPs in this model drastically attenuated tumor growth [83]. In addition to these ENS-derived neurotransmitters, chemogenetic activation of ex-trinsic Trpv1+ neurons from the dorsal root ganglia (DRG) was recently shown to de-crease cecal and colonic and MΦs, suggesting neurotransmitter involvement in modu-lating MΦs steady-state in the gut compartment [84]. Indeed, these studies have started to illuminate the potential of neuroimmune interactions between the ENS and intestinal MΦs.

Examining the effect of neurotransmitters on resident macrophages outside of intestinal tissues allows us to appreciate the broader principles that may govern ENS-MΦ interactions (Figure 2, inset). While BMDMs are phenotypically different from tissue-resident MΦs, neurotransmitter modulation of BMDMs provides deeper insight into at least the subset of intestinal macrophages constitutively replaced by circulating monocytes [85]. BMDMs also express serotonin receptors, and 5-HT signaling drives anti-inflammatory transcriptional programming in both human and murine macrophages [86]. Similarly, exposure of BMDMs to VIPs in vitro promotes an anti-inflammatory profile in favor of decreasing pro-inflammatory cytokine production [87]. CGRP, when ectopically introduced to LPS-activated BMDMs, increases the production of IL-10 [88]. The ability of these neurotransmitters released by the ENS to promote profound anti-inflammatory effects in hematopoietic macrophages warrants further exploration in the context of the gut.

Studies elucidating the effects of neurotransmitters on other tissue-resident MΦs further underscore these potential neuro–immune axes. For example, Kupffer cells (liver resident macrophages) respond to CGRP and decrease TNFα and IFNγ release in hepatitis models. Conditional ablation of the endogenous CGRP receptor Ramp1 further increased immune infiltration to the liver, exacerbated liver tissue damage, and decreased survival in murine hepatitis models [89]. Likewise, following skin and muscle injury, CGRP signaling to local macrophages bolsters efferocytosis, phagocytosis, and wound repair, and polarizes macrophages to an anti-inflammatory phenotype [90]. Serotonin signaling enhances TGFβ production of MΦs in the skin, supported by the observed decrease in fibrosis with the ablation of the serotonin receptor 5-HT7R [91]. Serotonin and VIP also increase phagocytosis of peritoneal MΦs, though the effect of the latter contrasts with a study illustrating VIPs’ inhibitory effect on peritoneal MΦ phagocytosis [92,93,94].

There is a great need to explore these neurotransmitter–MΦ interactions in the gut niche. There is a growing interest in understanding how neurotransmitters affect circulating MΦs and resident MΦs in a wide variety of tissues. However, MΦ phenotypes differ drastically across tissues [95]. For example, intestinal MΦs—despite being consistently exposed to food and commensal antigens—exhibit a remarkably anti-inflammatory identity. While they retain their highly bactericidal and phagocytic functionality, classic inflammatory stimuli (i.e., IFNγ, phorbol 12-myristate 13-acetate, and H. pylori urease) induce a minimal production of IL-1, IL-6, or TNFα by intestinal macrophages compared to blood monocytes [18]. It remains unclear how circulating monocytes, sensitive to inflammation, change to become anergic upon entering the intestinal niche. A deeper understanding of the molecular milieu of the intestinal MΦ microenvironment is key to discerning their unique characteristics. With a large body of evidence pointing to the anti-inflammatory effects of neurotransmitters, the local contributions of the ENS to this reprogramming pose an attractive hypothesis warranting further investigation.

One way to investigate the role of ENS-MΦ crosstalk in gut physiology is through the selective elimination of one or the other. The studies mentioned above reinforced the role of intestinal MΦs in the ENS control of bowel function. Orthogonal approaches using genetic strategies (e.g., Cx3cr1^−/−^) and pharmaceuticals (e.g., clodronate or the α-CSFR1 antibody) cause abnormal bowel motility, alter tissue healing as well as neuron survival in the face of infection, and impair normal ENS development. However, the effect of ENS ablation on MΦs remains a mystery. Utilizing diseases where the ENS is compromised (either by degeneration or a failure of ENS precursors to colonize the bowel embryologically) may provide insight into how the absence of ENS signaling alters gut MΦ phenotypes.

Achalasia is a rare neurodegenerative disorder characterized by the loss of myenteric inhibitory neurons, conferring dysphagia. Interestingly, histological analysis of mucosal tissue from patients with achalasia reveals a skewing of MΦs towards the M1 phenotype [96]. Altered MΦ function is further supported by the discovery of C1QC^+^ macrophages in the sequencing of human achalasia tissue samples, which exhibit a transcriptional profile similar to the microglia found in neurodegenerative microenvironments [97].

Research on Hirschsprung Disease (HSCR), a disease characterized by an incomplete migration of ENS precursors during fetal development, causing distal colon aganglionosis, also suggests that macrophages are perturbed in the absence of ENS signaling. Pro-inflammatory M1 MΦs infiltrate the colon in human patients with HSCR [98]. In human HSCR disease, decreased IL-23 production by MΦs in the aganglionic colon is associated with a better prognostic outcome [99]. In a HSCR mouse model, the short-term depletion of macrophages decreases enterocolitis risk severity [98]. While these studies allude to the importance of ENS-MΦ interactions in the clinical context of achalasia and HSCR, disentangling these observations from other contributing factors such as extrinsic neuron innervation, dysbiosis, and altered epithelial biology, remains incomplete.

## 4. Conclusions and Perspectives

The ENS underpins diverse aspects of intestinal function. Research accumulated over the last three decades suggests an important role for neuroimmune interactions in gut homeostasis, mucosal immunity, and intestinal inflammation. In this review, we spotlighted intestinal macrophages as potential executioners of ENS–immune signaling, focusing on existing co-dependencies of MΦs and the ENS in the context of development and infection. Indeed, many constituents of the ENS have the potential to affect macrophage polarization, cytokine production, and the phagocytic index in tissues outside the gut.

The intricate interplay between ENS and intestinal MΦs in the context of disease underscores the pivotal role that MΦs play in maintaining gut homeostasis and pathogenic defense. MΦs from diverse tissues exhibit responsiveness to neuropeptides and neurotransmitters released by the ENS, but still, many key interactions need to be explored. Are these neuroimmune interactions mirrored in the gut by the ENS? How is the ENS-MΦ axis established during development and throughout weaning, both from a spatial and temporal perspective? How does the ENS contribute to the unique anti-inflammatory MΦ programming seen at homeostasis? To interrogate these questions is to pave the way for novel prophylactic and therapeutic treatments to treat and prevent inflammatory intestinal diseases.

## Figures and Tables

**Figure 1 biomolecules-14-01103-f001:**
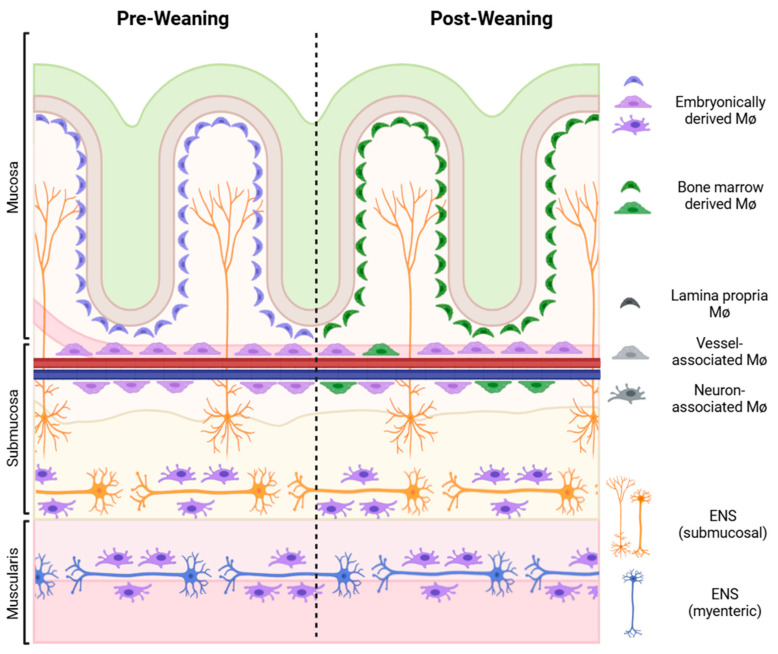
ENS–macrophage niche. MΦs are in close proximity to their targets: the epithelium, blood vessels, and the enteric nervous system, particularly the submucosal plexus. The MΦ population shifts after weaning from predominant embryonic yolk-sac and fetal-liver-derived cells to hematopoietic bone-marrow-derived cells.

**Figure 2 biomolecules-14-01103-f002:**
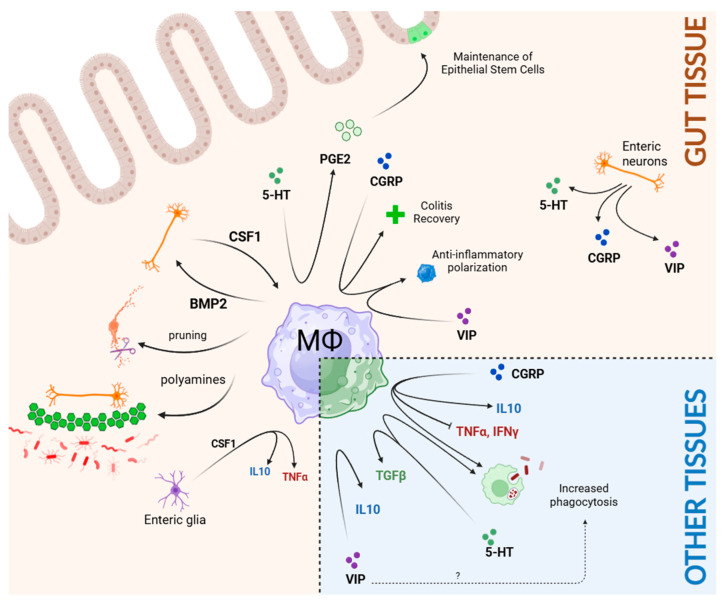
MΦ neuroimmune interactions. MΦs are under the influence of enteric neurons (orange) both during homeostasis and in the setting of inflammation. Common neurotransmitters and neuropeptides produced by the ENS have profound effects on MΦs found in gut tissues and other non-gut tissues (inset, bottom right).

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
