# Peer review of "Neuroimmune Crossroads: The Interplay of the Enteric Nervous System and Intestinal Macrophages in Gut Homeostasis and Disease"

_biomolecules, 2024, doi:10.3390/biom14091103_

Round 1

Reviewer 1 Report

Comments and Suggestions for Authors

The interplay between the enteric nervous system (ENS) and intestinal macrophages is a vital component of gut homeostasis and immune regulation. The ENS, often termed the "second brain," is a complex network of neurons embedded within the gut wall, responsible for controlling various gastrointestinal functions. Intestinal macrophages, as resident immune cells, play a crucial role in balancing immune tolerance and inflammation in the gut. However, our understanding of this interplay is still in its early stages.

This mini-review effectively summarizes the current progress in understanding the interactions between the ENS and intestinal macrophages in maintaining gut homeostasis and their roles in disease. The insights provided will likely stimulate further interest and research in this emerging field. The review is well-written and well-organized. My only suggestion is to include more discussion on the development of gut monocytes and how the ENS may interact differently with these two distinct myeloid populations, as well as how these interactions may reciprocally influence one another.

Author Response

Comment: “include more discussion on the development of gut monocytes and how the ENS may interact differently with these two distinct myeloid populations, as well as how these interactions may reciprocally influence one another.”

Response: Thank you for the constructive comments and thank you for the suggestion; we agree that a stronger emphasis on the different types of macrophages in the intestines may be helpful in framing the foundation of the review. We have included additional references that illustrate the complexity of the heterogeneity of the intestinal macrophages (lines 182-183). Unfortunately, much of the studies of ENS-macrophage interactions highlighted in the review don’t distinguish between the two subtypes, as the nomenclature is relatively more recent. However, we provided clarifications of the existing studies (lines 224-225) and made a point to communicate the need to delve into this discrepancy (lines 225-227).

Reviewer 2 Report

Comments and Suggestions for Authors

The manuscript written by Lou M., Heuckeroth R.O., and Butler Tjaden N.E provides an insightful and comprehensive review of the neuroimmune crosstalk between the enteric nervous system and intestinal macrophages. The authors have done an excellent job detailing the role of macrophages in maintaining gut homeostasis and their importance in various pathological contexts. I particularly appreciate the integration of interaction between enteric neurons and macrophages and the inclusion of enteric glia in this communication. The review efficiently incorporates recent literature.

A minor suggestion: A recently published paper could add nuance to the discussion in the section on CGRP release from the ENS and control of macrophages. This new study highlights the influence of TRPV1-expressing DRG neurons in regulating immune recruitment within the intestine through CGRP release (PMID: 39088603). Although the author’s manuscript was submitted prior to the publication of this study, its incorporation could enrich the discussion by providing additional perspectives on the complex interactions between ENS, intestinal immunity, and peripheral afference.

Author Response

Comment: “Although the author’s manuscript was submitted prior to the publication of this study, its incorporation could enrich the discussion by providing additional perspectives on the complex interactions between ENS, intestinal immunity, and peripheral afference.”

Response: Thank you for the astute suggestion on the inclusion of this literature; we wholeheartedly agree that it provides an additional facet to ENS-macrophage interactions and have included the reference in our discussion on neurotransmitter effect on intestinal macrophages (lines 262-264).

Reviewer 3 Report

Comments and Suggestions for Authors

Dear authors,

Thank you so much for submitting your great work. The article represents the review with a focus on cross talk between macrophages and neuron. The article is sufficiently novel and very interesting to warrant publication. All the key elements are presented and described clearly. There are some minor issues as below...

1.    Line 96-97: the description lacks reference(s). Please add enough refs for non-expert for neurons.

2.    Some sentences sound long. i.e. Line 39-43: Shorter sentence would be more readable. Please clarify it.

3.    Figure needs legend: I could not read the Fig1 legend.

4.    There are some errors in grammars and format. Please check through all the text, tables and figures. For example.....

a.    Line 06 and 12: The mark of corresponding author is deferent. + vs. *

b.    Line 89: enteric nervous system (ENS) ==> ENS

c.    Line 122: Spell out NMU

d.    Line 130 and 132: Please keep consistence. IL-6 vs. IL6.

e.    Line 132: Please keep consistence. Rorgt vs. RORγt.

f.      Line 132: + should be uppercase

g.    Line 139: Spell out S. typhi upon first appearance.

h.    Line 150: Macrophages (Mφ) ==> Mφ

i.      Line 160: + should be uppercase

j.      Line 169 and 170: Please keep consistence. gut Mφ vs. intestinal Mφ

k.     Line 199: remove extra space? after “models display”

l.      Line 256: Please keep consistence. Tgf vs. TGF

Author Response

Comment 1: “ Line 96-97: the description lacks reference(s). Please add enough refs for non-expert for neurons.”

Response 1: We agree with this comment and have added additional references for the anatomy and composition of the ENS (lines 97, 101).

Comment 2: “Some sentences sound long. i.e. Line 39-43: Shorter sentence would be more readable. Please clarify it.”

Response 2: We agree and have changed the sentence for better clarity and readability (lines 42-43)

Comment 3: “ Figure needs legend: I could not read the Fig1 legend.”

Response 3: Apologies, we have adjusted the scaling for Fig1’s figure legend

Comment 4: “some errors in grammars and format”

Response 4: Thank you for the keen eye! We have gone through and fixed the inconsistencies (lines 89, 122, 131-132, 139-140, 150, 169, 259). We have also adjusted the cytokine names to be more consistent eg. IL-6.